# Comparative assessment of the feasibility and validity of daily activity space in urban and non-urban settings

Sarah M. Kwiatek[1], Liang Cai[2], Kathleen A. Cagney[1], William E. Copeland[3], V. Joseph Hotz[4], Rick H. Hoyle[5]*

1 Institute for Social Research, University of Michigan, Ann Arbor, Michigan, United States of America, 2 Department of Sociology, University of Chicago, Chicago, Illinois, United States of America, 3 Department of Psychiatry, University of Vermont, Burlington, Vermont, United States of America, 4 Department of Economics, Duke University, Durham, North Carolina, United States of America, 5 Department of Psychology & Neuroscience, Duke University, Durham, North Carolina, United States of America

* rhoyle@duke.edu

**Data Availability Statement:** All data can be accessed in files associated with the Rural Activity Space project at the Open Science Framework (https://osf.io/3rgvb/).

## Abstract

Activity space research explores the behavioral impact of the spaces people move through in daily life. This research has focused on urban settings, devoting little attention to non-urban settings. We examined the validity of the activity space method, comparing feasibility and data quality in urban and non-urban contexts. Overall, we found that the method is easily implemented in both settings. We also found location data quality was comparable across residential and activity space settings. The major differences in GPS (Global Positioning System) density and accuracy came from the operating system (iOS versus Android) of the device used. The GPS-derived locations showed high agreement with participants' self-reported locations. We further validated GPS data by comparing at-home time allocation with the American Time Use Survey. This study suggests that it is possible to collect daily activity space data in non-urban settings that are of comparable quality to data from urban settings.

## Introduction

This study assessed the feasibility of research on "activity space"—the geographic area through which people move in everyday life—in non-urban settings. The concept of activity space is rooted in geographic research and refers to the spatial patterns of routine activity wherein regular events occur with distinctive rhythms, tempos, and timings [1,2]. It is a spatiotemporal construct that captures the places people encounter during their activities in everyday life [3,4]. People may travel out of and potentially far from their residential space for errands, obligations, work, or social engagements. Thus, activity spaces include residential areas but also extend beyond them. The distance people travel, as well as the spaces they visit, vary in terms of geographic, structural, cultural, and social features. Intuitively, urban and non-urban spaces are likely to differ because of these features; however, a recent review of urban mobility

**Funding:** This work was funded by the National Institute on Drug Abuse Grant P30 DA023026 (RHH) and the National Institute on Aging Grants P30 AG034424 (VJH) and P30 AG066619 (KAG). The funders had no role in study design, data collection and analysis, decision to publish, or preparation of the manuscript.

**Competing interests:** The authors have declared that no competing interests exist.

research noted that simultaneous comparisons of these activity spaces are currently lacking [1]. Activity space studies are typically conducted over one week, during which participants must keep a study phone (or wearable device) on them at all times so that GPS (Global Positioning System) data can be continuously recorded. Studies may also include one or more daily surveys to collect contextual information on participants' movements and validate their GPS data. Past research indicates that multi-method approaches that combine GPS and self-report data are particularly advantageous for accurately characterizing a person's activity space [5]. In particular, studies that include ecological momentary assessment (EMA) data, which refers to data collected via multiple daily surveys, provide detailed and specific contextual information on a person's activity space [6].

Previous research in urban contexts has successfully implemented these methods given that urban areas typically have ample cell coverage and participants are often close to research facilities should they have device-related or payment issues [7]. In non-urban contexts, more limited cellular coverage could disrupt data collection. Participants may also have to travel long distances to facilities and may be less inclined to agree to such tracking. In prior urban studies of activity space, some additional challenges have been identified, such as the possibility that tracking may lead participants to alter their behavior, the potential for one week to be insufficient for identifying behavioral patterns, and the broader issue of being unable to capture the heterogenous experiences of people in different neighborhood and social contexts [1]. Despite these challenges, each study adds to the emerging activity space literature and is crucial for addressing these limitations. We implemented this method in non-urban contexts to address the lack of activity space research within non-urban areas [1] and identify potential implementation challenges.

Extending these methods to non-urban spaces allows for fundamental comparative work across urban and non-urban areas and unprecedented documentation of the form and span of activity space in non-urban contexts. Non-urban residents likely differ from their urban counterparts in daily activities and their distance from resources and amenities (see [8]). Highlighting such differences is critical for future research on the links between activity space and long-term outcomes such as health, well-being, and economic opportunity. To that end, the present study extended activity space concepts and methods in several important ways.

First, we examined the feasibility and reliability of using activity space methods in non-urban settings. Drawing on protocols and software used in the Activity Space, Social Interaction, and Health Trajectories in Later Life study in urban neighborhoods of Chicago [9], we were able to assess the relative cost and quality of using the activity space method in both settings.

Second, unlike the Chicagor activity space study, we evaluated the feasibility of a fully remote implementation of protocols. The study was conducted with no face-to-face interaction and minimal remote interaction between the research staff and participants. Nearly all interactions were handled by telephone, email, or text message. This approach was necessitated, in part, because this assessment was conducted during the COVID-19 pandemic, which permitted no in-person contact between participants and field staff.

Third, unlike the Chicago study and other previous activity space studies that provided participants with study-specific phones [10], our participants used their personal mobile devices. This required installation of the application used for data collection and location tracking on both iPhones running iOS and Android phones. Though this approach brought challenges, it eliminated having to provide participants with devices, the need to potentially carry two devices—their own and the study-provided one—and use of devices unfamiliar to them. It also offered an opportunity to compare the two operating systems.

To investigate the feasibility of the activity space method and the validity of the data it produces, we addressed two research questions:

1. How comparable are the challenges, costs, and data quality in applications of activity space methods between urban and non-urban settings? (Implementation)

2. How reliable are the methods implemented for capturing the daily locations and movement of people, and does reliability differ based on location and device type? (Reliability)

## Methods

### Sample

Participants were recruited from an ongoing longitudinal study in North Carolina, Research on Adaptive Interests, Skills, and Environments (RAISE), conducted by the Center for the Study of Adolescent Risk and Resilience (C-StARR) at Duke University ($N$ = 162; 92.6% Female, 7.4% Male; 72.8% White, 21.6% Black, 1.9% American Indian, 2.5% Asian, 3.1% Other). The RAISE sample consists of parent and child dyads. Parents between 36 and 50 years of age ($M$ = 44.93, $SD$ = 3.92) with smartphones were eligible to participate.

### Recruitment

The sample was divided into urban and non-urban groups based on the Rural-Urban Commuting Area Codes (RUCA) linked to the county of their home address (Urban: 1–3; Non-urban: 4–10) [11]. The initial plan was to recruit 20 urban and 20 non-urban participants; however, based on high response rates and positive feedback, recruitment continued beyond this initial target (Urban $N$ = 122; Non-urban $N$ = 40). Recruitment began 9 March, 2021 and continued until 11 June, 2021. The Duke University Campus Institutional Review Board reviewed and approved the study (Protocol 2021–0249).

Invitation emails were sent at the start of the week, followed by three email reminders across two weeks. The invitation email contained information about the study, eligibility criteria, and a link to the study registration form in Qualtrics. Participants who registered for the study were presented an IRB-approved informed consent form and indicated their consent to participate by typing their name and selecting "Next" to continue to the initial survey. Of the 377 participants contacted, 188 (49.9%) consented to be in the study. Of those who consented, 170 participants were eligible (15 were over the age of 50, two were under the age of 35, and 1 did not have a smartphone). Of those who consented and were eligible, 162 participants completed the study (two opted out mid-study for personal reasons/lack of time, and six participants either never downloaded the MetricWire app or downloaded the app but never finished the study).

### MetricWire

The survey instruments and software for monitoring movement were comparable to those in the Chicago activity space study. We used MetricWire, a mobile phone-enabled survey platform, to delivery surveys and record spatial movement. Participants downloaded the app on their personal devices through the Google Play Store or the Apple App store. All participants used MetricWire version 4.9 (with some variation between those who had 4.9.1 to 4.9.4). The MetricWire survey system is well-suited for daily survey studies. The app also retrieved detailed location data with little effort from participants beyond the initial app setup.

### Enrollment

Participants received enrollment information within 24 hours of providing consent. They received informative guides (available at https://osf.io/3rgvb/) with instructions to set up the app, including how to adjust their smartphone settings to allow the app to access their location data and notification settings. Though remote enrollment was occasionally challenging, most participants had no issue setting up their devices. Recruiters were also available to answer participants' questions by email, phone, or text.

### Study protocols

Over seven consecutive days, participants completed brief surveys in the morning, afternoon, and evening while the MetricWire app passively collected their real-time geo-coordinates. On the eighth day, participants completed an end-of-study survey to summarize their experiences during the past week. Participants received $5 for each daily survey, $10 for the final survey, and a $20 bonus if they completed 15 or more daily surveys for a total possible compensation of $135. Over the course of the study, the research team monitored and addressed issues with participant enrollment, completion of daily surveys, and location sensors. Issues with daily surveys or location sensors were identified either by MetricWire dashboard alerts or direct participant contact.

### Measures

For this report, we focus on the accuracy and validity of the geolocation data. The daily surveys served as validity measures via questions about where participants were throughout the day, which we compared with their GPS location. The surveys also provided contextual specificity about the characteristics of participants' activity space that will be used in later analyses. Finally, the surveys ensured participants had their phones turned on and near them at all times, a necessary feature for location data collection.

## Results

### Implementation

The implementation of comparable protocols for measuring activity spaces in urban and non-urban settings was successful. Participants found the MetricWire app easy to navigate and had few issues setting up the app or completing the surveys. Participants could also download the app on their personal devices regardless of their phone's operating system.

To enable location sensing, participants only needed to turn on a setting on their smartphone to allow the MetricWire app to access their GPS systems—the rest was automated. This feature allowed us to implement the study protocols remotely and ultimately resulted in an impressive amount of location data for participants in urban and non-urban areas. Overall, participants were highly compliant, with an average total survey completion rate of 86% for urban and 85% for non-urban residents. Participants reported, anecdotally, positive experiences with the study and its requirements (e.g., "It is a great app for surveys," "It was super easy," "Process was simple & directions provided were thorough and accurate.").

In terms of cost, the average compensation amounts were $117 for urban residents and $115 for non-urban residents. Most participants earned both the daily and final survey bonuses. Approximately 85% of urban and non-urban residents received the $20 daily survey bonus. Nearly 99% of urban and 98% of non-urban residents received the $10 final survey bonus. The total compensation distributed was $18,870. Though budget constraints may not allow replication of this compensation structure, we recommend including bonus incentives

for participants to maximize compliance rates. We also recommend offering several forms of payment. In the present study, participants could receive compensation via an Amazon gift card, a Target gift card, or a Food Lion (North Carolina grocery store chain) gift card. Among urban residents, 80% chose an Amazon gift card, 10% chose a Target gift card, and 10% chose a Food Lion gift card. Among non-urban residents, 75% chose an Amazon gift card, 10% chose a Target gift card, and 15% chose a Food Lion gift card.

Although the implementation was successful, we encountered several challenges that required intervention or assistance from the research staff. About 57% of urban and 45% of non-urban participants had one or more issues during the study. Common issues included the MetricWire app not being installed, location data not uploading, and missing a day of surveys. Problems with the location data often resulted from participant error (improper settings), with a few occasional app-related issues that MetricWire team quickly addressed. Most concerns categorized as "other" were participant-specific (e.g., needing to change their study timeline because of an upcoming vacation). Participants overwhelmingly preferred troubleshooting issues via text rather than by email or phone call.

The research team found during testing that several available sensors in MetricWire (e.g., geolocation, pedometer, and accelerometer) did not respond effectively. The use of multiple sensors and the length of the surveys (~90 items/survey) caused the app to crash during survey completion. The MetricWire support staff suggested choosing only one sensor to resolve this issue, so the team decided to use only the location sensor. For studies requiring multiple sensors, this may pose a challenge. Though we encountered a few other challenges during the study period, the majority of issues were encountered by the research team and not by participants.

## Reliability

To examine how reliable our GPS data were across settings (urban vs. non-urban) and device types (Android vs. iOS), we analyzed a series of data quality metrics, including geographic and time resolution, GPS accuracy, temporal coverage, the number of participation days, and the number of GPS points recorded. In particular, we inspected outliers for time gap, distance gap, and accuracy. We operationalized distance and time gaps as the gaps between pairs of consecutive GPS points from the same participant. We considered participation days as those with at least one recorded GPS point. GPS accuracy is a standard measure of certainty associated with GPS points [12]. GPS accuracy indicates the radius in meters associated with a 68% (or one standard deviation) confidence region around the GPS estimate. Higher values indicate more uncertainty around the estimate. Accuracy is subject to hardware and connectivity. The best location data are generated when participants allow GPS tracking and enable cellular and/or Wi-Fi data. Accuracy values are also influenced by environmental interference (e.g., high-rise buildings, atmospheric conditions), cellular coverage, and Wi-Fi coverage. Temporal coverage represented the total time location tracking was recorded during the study period. For example, 100% temporal coverage would mean the participant reported GPS data every minute during the study period. We also validated a subset of GPS location data for which participants' self-reported locations were available from the daily EMA surveys. Finally, we compared the estimate of the proportion of time spent at home to that from the American Time Use Survey [13].

During the seven-day study period, the MetricWire app passively and continuously recorded geolocation unless the device was off or the location tracking service was disabled. Locations may have also been recorded before or after the study period while the app was installed. To address this, we first filtered out all GPS points outside the study period. We further identified GPS points that appeared to be outliers, likely due to errors in the GPS software

**Table 1. Summary statistics for GPS quality metrics by participants' residential location and device.**

| | Location of residence[†] | | Device | |
|---|---|---|---|---|
| | **Urban** | **Non-Urban** | **Android** | **iOS** |
| Time gap (second)* | | | | |
| Mean | 14.438 | 11.675 | 32.911 | 2.783 |
| SD | 35.23 | 25.157 | 48.734 | 5.683 |
| Median | 2.597 | 1.816 | 5.859 | 1.842 |
| Distance gap (meter)* | | | | |
| Mean | 4.108 | 4.930 | 6.980 | 2.782 |
| SD | 4.259 | 4.426 | 4.055 | 3.657 |
| Median | 2.679 | 3.242 | 5.961 | 1.879 |
| Accuracy (meter)* | | | | |
| Mean | 16.475 | 18.659 | 24.248 | 12.870 |
| SD | 29.217 | 40.107 | 32.947 | 31.052 |
| Median | 11.409 | 11.777 | 19.114 | 7.133 |
| Number of days participated | | | | |
| Mean | 6.918 | 6.850 | 6.898 | 6.903 |
| SD | 0.508 | 0.949 | 0.662 | 0.634 |
| Median | 7 | 7 | 7 | 7 |
| Temporal coverage | | | | |
| Mean | 0.490 | 0.506 | 0.394 | 0.552 |
| SD | 0.247 | 0.295 | 0.292 | 0.219 |
| Median | 0.514 | 0.553 | 0.466 | 0.545 |
| Number of GPS points | | | | |
| Mean | 233,129 | 261,197 | 125,779 | 302,520 |
| SD | 148,804 | 165,816 | 126,013 | 126,893 |
| Median | 217,937 | 298,772 | 99,592 | 316,556 |
| % iOS users | 62.3% | 67.5% | - | - |
| N (participants) | 122 | 40 | 59 | 103 |

[†] Location of participant's residence.

* Calculated at participant level.

or cell towers. Specifically, we filtered out GPS points that produced distance gaps of more than three standard deviations of the observed distribution of these gaps (i.e., 9865.762 meters). This step removed 7,074 GPS points. We further dropped 9,771 GPS points associated with extremely high movement speed. We used 250 meters per second as the threshold (i.e., the high end of commercial airplane speed) as GPS points beyond that speed are likely erroneous. These steps resulted in 38,889,564 GPS points from the 162 participants in our analyses.

Table 1 presents summary statistics for these measures by residential location and device type. Although more participants used iOS than Android devices, the split is similar between urban (62.3% iOS users) and non-urban residents (67.5% iOS users). Across the various quality and coverage measures, we found virtually no difference between urban and non-urban residents. The differences in median values of time gaps, distance gaps, and the number of participation days was much smaller across urban and non-urban locations than differences in mean values, which were influenced by a small number of high values on these metrics. However, major differences were observed between Android and iOS devices in all of the GPS quality metrics except for the number of participation days, with Android devices reporting GPS data of lower quality and coverage. With iOS devices, GPS points were recorded more

**Table 2. Summary statistics for GPS quality metrics and incidence of outliers by participants' residential and activity space locations.**

| | Urban residence[†] | | Non-Urban residence[†] | |
|---|---|---|---|---|
| | **Urban location*** | **Non-Urban location*** | **Urban location*** | **Non-Urban location*** |
| Time gap | | | | |
| Mean | 2.402 | 1.790 | 1.299 | 2.143 |
| SD | 163.344 | 67.931 | 42.820 | 154.762 |
| Median | 1 | 1 | 1 | 1 |
| % outlier (>1 hour) | 0.006% | 0.003% | 0.002% | 0.005% |
| Distance gap | | | | |
| Mean | 2.634 | 9.574 | 9.692 | 3.711 |
| SD | 39.385 | 86.570 | 54.731 | 33.639 |
| Median | 0 | 0.453 | 0.584 | 0 |
| % outlier (>1 km) | 0.009% | 0.028% | 0.026% | 0.008% |
| Accuracy | | | | |
| Mean | 10.881 | 10.550 | 12.303 | 11.452 |
| SD | 227.764 | 99.989 | 46.622 | 42.535 |
| Median | 4.85 | 5 | 5.07 | 4.80 |
| % outlier (>1 km) | 0.017% | 0.054% | 0.033% | 0.017% |
| N (GPS points) | 27,530,583 | 911,063 | 814,252 | 9,633,618 |
| % GPS points by location | 92.207% | 7.793% | 3.203% | 96.797% |

[†] Location of participant's residence.

* Location where GPS reading was taken.

frequently and accurately than Android devices, as evidenced by iOS devices' lower average time and distance gaps. The average coverage rate across the participation period was also higher among iOS users than Android users.

To better understand how outliers affected differences in the GPS quality measures, we disaggregated the results in Table 1 by location and device type and added results for the share of outliers in the distribution of GPS points. Table 2 presents statistics for the quality measures and outliers by where the GPS points were recorded as well as the participant's residential location. We used 1 km as the threshold for outliers in the distance gap and accuracy metrics and 1 hour for the time gap. Across device types, urban residents were predominantly in urban settings (over 92% of GPS points) and non-urban residents in non-urban settings (over 97% of GPS points).

As shown in Table 2, the incidence of outliers for the various GPS quality metrics was small for both residential and activity space locations. For time gaps, only 0.001 to 0.007% of GPS points were considered outliers across different location types. For distance gaps, 0.008 to 0.040% of GPS points were separated by more than 1 km. The highest share of distance gap outliers was among urban residents' non-urban activity spaces and non-urban residents' urban activity spaces. The percent of GPS points with poor accuracy ranged between 0.017 and 0.140%, with higher rates of inaccuracy coming from participants' activity spaces in locations different from their residential settings.

We also examined the distribution of outliers for the three quality metrics by device type. S1 and S2 Tables present the statistics comparable to those in Table 2 by Android and iOS devices, respectively. We found that Android devices generally performed worse than iOS devices across all combinations of activity space and residential locations. Again, we see higher rates of inaccuracy coming from participants' activity spaces in locations different from their residential setting.

**Table 3. Comparison between EMA-reported location and GPS-derived location.**

| | Location of Residence | | | | Device | | | |
|---|---|---|---|---|---|---|---|---|
| | Urban | | Non-urban | | Android | | iOS | |
| EMA GPS | At home | Not home | At home | Not home | At home | Not home | At home | Not home |
| 20m | | | | | | | | |
| At home | 927 | 91 | 286 | 29 | 506 | 36 | 707 | 84 |
| | 89.652% | 13.788% | 81.948% | 11.111% | 84.474% | 10.557% | 90.179% | 14.483% |
| Not home | 107 | 569 | 63 | 232 | 93 | 305 | 77 | 496 |
| | 10.348% | 86.212% | 18.052% | 88.889% | 15.526% | 89.443% | 9.821% | 85.517% |
| 50m | | | | | | | | |
| At home | 966 | 101 | 319 | 34 | 551 | 43 | 734 | 92 |
| | 93.424% | 15.303% | 91.404% | 13.027% | 91.987% | 12.610% | 93.622% | 15.862% |
| Not home | 68 | 559 | 30 | 227 | 48 | 298 | 50 | 488 |
| | 6.576% | 84.697% | 8.596% | 86.973% | 8.013% | 87.390% | 6.378% | 84.138% |
| 100m | | | | | | | | |
| At home | 974 | 103 | 342 | 35 | 564 | 45 | 752 | 93 |
| | 94.197% | 15.606% | 97.994% | 13.410% | 94.157% | 13.197% | 95.918% | 16.035% |
| Not home | 60 | 557 | 7 | 226 | 35 | 296 | 32 | 487 |
| | 5.803% | 84.394% | 2.006% | 86.590% | 5.843% | 86.804% | 4.082% | 83.966% |
| Not matched | 255 | 184 | 75 | 78 | 104 | 58 | 226 | 204 |

Unit of analysis in this table is EMA observations. Each participant is asked to respond to EMA surveys 3 times a day for 7 days, with an average of 17.877 EMAs completed (sd = 3.106).

Next, we compared participants' self-reported locations in the daily EMA surveys to participants' GPS-derived locations. For each EMA observation, we found the GPS location passively recorded at the same timestamp and examined if the locations aligned (Table 3). In the GPS-recorded location data, we used the building footprint and its extensions to delineate home, including 20m, 50m, and 100m buffers around the home building footprint. As seen in Table 3, most of the matched cases (for which we have a valid GPS point for the minute of the survey) had GPS and EMA both reporting at-home status. As we increased the buffer size around the home building footprint, the alignment between the two sources increased, reaching more than 94% agreement with the 100m buffer. When participants indicated being someplace other than home, our GPS-based results located participants to non-home spaces at a high likelihood (over 83% agreement).

Finally, using the GPS data and the various definitions of home location, we compared the proportion of time spent at home by participants in our study with comparable measures in the 2021 ATUS data for metro and non-metro samples. We use ATUS's definition of metro and non-metro instead of grouping participants into urban and non-urban based on their residential counties because participants' residential counties are only identified for 44.52% of the ATUS sample. Metro participants are those from metropolitan statistical areas (MSAs) per 2010 Decennial Census definitions. Fig 1 compares 95% confidence intervals derived from the ATUS data for the proportion of wake time ATUS participants reported spending at home with the corresponding intervals based on the data in our samples, using various definitions of what constituted being at home for our participants. The dark gray bar represents the 95% confidence interval derived from the ATUS metro sample aged 35 to 50. The light gray bar represents that of the ATUS non-metro sample of the same age range. For the ATUS metro sample ($N$ = 2,078), the average wake time spent at home was 63.43%, with a standard deviation of

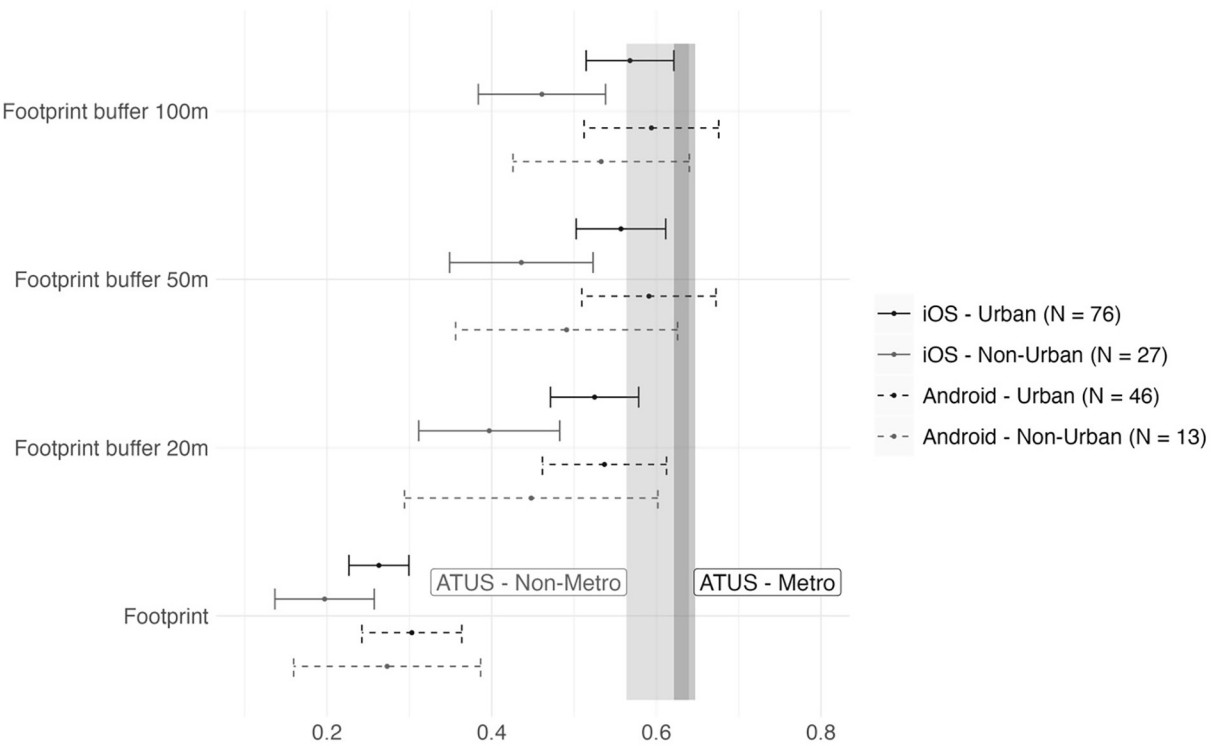

The shaded areas represent the 95% confidence intervals from the ATUS data and the bars display the corresponding intervals for various subgroups in the RAISE data.

**Fig 1. Comparison of proportion of wake time (8AM– 8:30PM) spent at home between the American Time Use Survey sample and our sample.**

29.88 and a median of 67.56%. For the non-metro sample, the average time spent at home was 60.16%, with a standard deviation of 32.11 and a median of 58.09%). The bars in Fig 1 display the corresponding intervals for various subgroups of our data. Note that confidence intervals for our sample were wider than those for the ATUS sample due to our smaller sample sizes.

According to Fig 1, the distributions of the proportion of wake time (here defined as 8:00 AM– 8:30 PM) for our participants was most similar to that from ATUS when using 50m and 100m buffers around the home building footprint as the definition of being at home. Expanding the buffer size from 50m to 100m only marginally increased the average proportion of time at home, particularly among urban residents, indicating that the 50m buffer captured most of the at-home activities. Like the ATUS results, urban residents in our sample spent more time at home than non-urban residents, especially iOS users. This finding could be due to the study being conducted during the COVID-19 pandemic, when the high population density in urban areas could mean that leaving home represented a higher risk of contracting the virus.

## Discussion

Our findings indicate that activity space research is feasible in non-urban settings and comparable in participant experience and data quality to activity space research in urban settings. The results also highlight the feasibility of conducting activity space research via remote protocols. These findings will allow future researchers to expand this method to contexts and communities that have yet to be studied and to samples that may have previously been inaccessible. Such advancements are critical in understanding and supporting communities that might

otherwise be overlooked due to the challenges associated with tracking movement in their locations.

## Limitations and future directions

This study was conducted during the COVID-19 pandemic (March 2, 2021, to June 11, 2021), when many people were required to adhere to governmental restrictions related to travel and mobility. These restrictions limit the generalizability of the specific findings. However, the results of this feasibility evaluation are encouraging and suggest that a method of this kind can be effective when in-person administration is not feasible.

It is also necessary to consider that participants were recruited from an existing sample with an established trusting relationship with the research team. It is likely that this eased the recruitment burden by reducing possible hesitation regarding the methods used in this study. Future researchers that hope to conduct similar studies with samples they have not previously studied will need to carefully consider recruitment plans and prepare to establish trust with participants in order to manage concerns about passive location tracking.

Although we employ several metrics to assess GPS data quality, our assessment is not comprehensive. There were technical issues in our GPS data collection that were beyond the control of researchers. For example, we observed a few cases of GPS jumps where two temporally consecutive GPS points from the same device were separated by huge distance gaps. In the present analysis, we applied some simple rules based on the speed and distribution of distance gaps to filter out highly unrealistic points. Future assessments could look into the technical side of the problem to identify the reasons behind such jumps.

Finally, we identified some limitations from an implementation perspective. In future analyses, investigators may wish to explore additional ways to assess reliability and validity. Longitudinal data, for instance, would allow for a range of evaluation efforts that could speak to individual life-course changes alongside changes in context. Other analyses may explore how the GPS and EMA components relate to other types of data collection (e.g., traditional social survey) and whether this augmentation helps or hinders securing additional data.

Our findings have important implications for comparative work and research in locations where other data collection methods might not be logistically or monetarily possible. The challenges we have identified appear to be addressable, and technological advancements in the near future will likely improve the quality of measurement of participants' activity spaces, regardless of locational settings or devices used to record them. Our results suggest great promise in the use of activity space approaches for a wide range of populations and contexts and for the integration of such work with other forms of data collection.

## Supporting information

**S1 Table. Summary statistics for Android devices only.** GPS quality metrics and incidence of outliers by participants' residential and activity space locations.
(DOCX)

**S2 Table. Summary statistics for iOS devices only.** GPS quality metrics and incidence of outliers by participants' residential and activity space locations.
(DOCX)

## Acknowledgments

We are grateful to Erin Davisson and Jenny Park for help with sampling and recruitment and Luke Pomrenke for help with recruitment and data collection.

## Author Contributions

**Conceptualization:** Kathleen A. Cagney, William E. Copeland, V. Joseph Hotz, Rick H. Hoyle.

**Data curation:** Sarah M. Kwiatek, Liang Cai.

**Formal analysis:** Sarah M. Kwiatek, Liang Cai.

**Funding acquisition:** Kathleen A. Cagney, William E. Copeland, V. Joseph Hotz, Rick H. Hoyle.

**Investigation:** Sarah M. Kwiatek.

**Methodology:** Kathleen A. Cagney, William E. Copeland, V. Joseph Hotz, Rick H. Hoyle.

**Project administration:** Rick H. Hoyle.

**Resources:** Rick H. Hoyle.

**Visualization:** Liang Cai.

**Writing – original draft:** Sarah M. Kwiatek.

**Writing – review & editing:** Liang Cai, Kathleen A. Cagney, William E. Copeland, V. Joseph Hotz, Rick H. Hoyle.

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
