## [Decision Letter · Decision Letter 0]

5 Dec 2023

PONE-D-23-31641Comparative assessment of the feasibility and validity of daily activity space in urban and non-urban settingsPLOS ONE

Dear Dr. Hoyle,

Thank you for submitting your manuscript to PLOS ONE. After careful consideration, we feel that it has merit but does not fully meet PLOS ONE’s publication criteria as it currently stands. Therefore, we invite you to submit a revised version of the manuscript that addresses the points raised during the review process.

We look forward to receiving your revised manuscript.

Kind regards,

Abel C.H. Chen

Academic Editor

PLOS ONE

“This work was funded by the National Institute on Drug Abuse Grant P30 DA023026 (RHH) and the National Institute on Aging Grants P30 AG034424 (VJH) and P30 AG066619 (KAG).”

3. We notice that your supplementary tables are included in the manuscript file. Please remove them and upload them with the file type 'Supporting Information'. Please ensure that each Supporting Information file has a legend listed in the manuscript after the references list.

Reviewers' comments:

Reviewer's Responses to Questions

**Comments to the Author**

1. Is the manuscript technically sound, and do the data support the conclusions?

Reviewer #1: Yes

Reviewer #2: Yes

2. Has the statistical analysis been performed appropriately and rigorously? 

Reviewer #1: Yes

Reviewer #2: Yes

3. Have the authors made all data underlying the findings in their manuscript fully available?

Reviewer #1: Yes

Reviewer #2: Yes

4. Is the manuscript presented in an intelligible fashion and written in standard English?

Reviewer #1: Yes

Reviewer #2: Yes

5. Review Comments to the Author

Reviewer #1: Many thanks for the opportunity to review this interesting submission which I really enjoyed reading. Your submission demonstrates an excellent standard of writing and was clear and easy to read and follow. You might want to further explain the phrase used on p 11 line 249 "the right tails in the metrics" which may be unfamiliar to some readers. I have no further suggestions to improve.

Reviewer #2: This is a useful report on how to record location+survey data from urban and rural settings remotely, to identify and analyze activity spaces. This approach can be useful for research in various fields. Congratulations for the clarity in the manuscript, and for providing open access to fieldwork protocols and data. Some minor comments:

1. Authors listed ‘intensive longitudinal design’ as a keyword; however, other than the fact that the data came from a larger longitudinal study, no reference to the “intensive longitudinal design” was made in the manuscript. I’d consider removing this from the list.

2. It could be useful to see a brief background of studies that have focused in urban studies and some if they exist who have analyzed areas beyond cities, and especially in rural settings. Especially, authors could provide a brief description of limitations/challenges identified in those studies, if any.

3. In table 1, why is the number of participation days so low, with values around 0.5? Also, why is temporal coverage median 7 – maybe these two variables are reversed in the table?

Thank you.

6. PLOS authors have the option to publish the peer review history of their article (what does this mean?). If published, this will include your full peer review and any attached files.

Reviewer #1: No

Reviewer #2: No

---

## [Author Response · Author response to Decision Letter 0]

21 Dec 2023

Please see Response to Reviewers document.

---

## [Decision Letter · Decision Letter 1]

8 Jan 2024

Comparative assessment of the feasibility and validity of daily activity space in urban and non-urban settings

PONE-D-23-31641R1

Dear Dr. Hoyle,

We’re pleased to inform you that your manuscript has been judged scientifically suitable for publication and will be formally accepted for publication once it meets all outstanding technical requirements.

Kind regards,

Abel C.H. Chen

Academic Editor

PLOS ONE

Additional Editor Comments (optional):

Reviewers' comments:

Reviewer's Responses to Questions

**Comments to the Author**

1. If the authors have adequately addressed your comments raised in a previous round of review and you feel that this manuscript is now acceptable for publication, you may indicate that here to bypass the “Comments to the Author” section, enter your conflict of interest statement in the “Confidential to Editor” section, and submit your "Accept" recommendation.

Reviewer #2: All comments have been addressed

2. Is the manuscript technically sound, and do the data support the conclusions?

Reviewer #2: Yes

3. Has the statistical analysis been performed appropriately and rigorously? 

Reviewer #2: Yes

4. Have the authors made all data underlying the findings in their manuscript fully available?

Reviewer #2: Yes

5. Is the manuscript presented in an intelligible fashion and written in standard English?

Reviewer #2: Yes

6. Review Comments to the Author

Reviewer #2: (No Response)

7. PLOS authors have the option to publish the peer review history of their article (what does this mean?). If published, this will include your full peer review and any attached files.

Reviewer #2: No

---

## [Editor Report · Acceptance letter]

16 Jan 2024

PONE-D-23-31641R1 

PLOS ONE

Dear Dr. Hoyle, 

I'm pleased to inform you that your manuscript has been deemed suitable for publication in PLOS ONE. Congratulations! Your manuscript is now being handed over to our production team.

Kind regards, 

on behalf of

Dr. Abel C.H. Chen 

Academic Editor

PLOS ONE